# Solute carrier family 2 member 2 (glucose transporter 2): a common factor of hepatocyte and hepatocellular carcinoma differentiation

Yejin Kim[1☉], Yu Yeuni[2☉], Hye Jin Heo[3☉], Eun Sun Kim[4], Kyungjae Myung[4], Ninib Baryawno[5], Yun Hak Kim [3,6*], Chang-Kyu Oh[1,7*]

1 Department of Convergence Medical Sciences, School of Medicine, Pusan National University, Yangsan, Republic of Korea, 2 Biomedical research institute, School of Medicine, Pusan National University, Yangsan, Republic of Korea, 3 Department of Anatomy, School of Medicine, Pusan National University, Yangsan, Republic of Korea, 4 Center for Genomic Integrity, Institute for Basic Science, Ulsan, Republic of Korea, 5 Childhood Cancer Research Unit, Department of Women's and Children's Health, Karolinska Institutet, Stockholm, Sweden, 6 Department of Biomedical Informatics, School of Medicine, Pusan National University, Yangsan, Republic of Korea, 7 Department of Biochemistry, School of Medicine, Pusan National University, Yangsan, Republic of Korea

☉ These authors contributed equally to this work.
* yunhak10510@pusan.ac.kr (YHK); ck1988@pusan.ac.kr (CKO)

## Abstract

GLUT2 (SLC2A2), a vital glucose transporter in liver, pancreas, and kidney tissues, regulates blood glucose levels and energy metabolism. Beyond its metabolic role, SLC2A2 contributes to cell differentiation and metabolic adaptation during embryogenesis and tissue regeneration. Despite its significance, the role of SLC2A2 in liver differentiation and hepatocellular carcinoma (HCC) remains underexplored. This study investigated SLC2A2's role in liver differentiation using in silico, in vitro, and in vivo approaches. Analysis of GEO datasets (GSE132606, GSE25417, GSE67848) and TCGA HCC data revealed that while SLC2A2 expression decreases with HCC progression, stemness-associated genes, including SOX2 and POU5F1, are upregulated. Zebrafish embryos injected with SLC2A2-targeting morpholino exhibited reduced expression of the liver differentiation marker *fabp10a* without significantly altering the hepatoblast marker *hhex*. In HepG2 cells, SLC2A2 knockdown increased stemness and IGF1R pathway markers, indicating a shift toward less differentiated states. These findings suggest that SLC2A2 supports liver differentiation by regulating glucose metabolism and suppressing pathways associated with stemness and malignancy. Targeting SLC2A2 may serve as a promising therapeutic strategy for liver-related diseases, particularly HCC, by addressing its dual role in differentiation and tumor progression. Further mechanistic studies are warranted to fully elucidate these processes.

**Data availability statement:** All GSE data files are available from the GEO database (GSE132606, GSE25417, GSE67848).

**Funding:** This work was supported by the Bio & Medical Technology Development Program of the National Research Foundation (NRF) funded by the Korean government (MSIT) (No. RS-2023-00223764 to CO, RS-2023-00207946 to YHK).

**Competing interests:** The authors have declared that no competing interests exist.

**Abbreviations:** IGF1R, insulin-like growth factor 1 receptor; *IGF2*, insulin-like growth factor 2; *SLC2A2*, solute carrier family 2 member 2; *GLUT*, glucose transporter; HCC, hepatocellular carcinoma; TCGA, The Cancer Genomic Atlas; WISH, whole-mount in situ hybridization; DIG, digoxigenin; qPCR, quantitative real-time polymerase chain reaction; dpf, days post-fertilization; RT-PCR, real-time polymerase chain reaction; BSA, bovine serum albumin; HepG2, human HCC cells.

## Introduction

Hepatocellular carcinoma (HCC) remains a significant global health challenge due to its high prevalence, poor prognosis, and the limited efficacy of current therapeutic options [1,2]. While traditional treatments such as surgery, chemotherapy, and targeted therapies have been implemented, their effectiveness is often compromised by the molecular heterogeneity of HCC and its aggressive nature [3,4]. Consequently, there is growing interest in the molecular classification of HCC to more effectively tailor treatments by targeting specific genetic alterations [5,6]. Recent advancements in this area emphasize the critical role of molecular classification, which categorizes tumors based on distinct genetic and molecular profiles, thereby identifying key genes and pathways as potential therapeutic targets [7,8]. This strategy paves the way for more precise and personalized treatment modalities [9]. Nevertheless, the prognosis for HCC patients remains dire, highlighting the necessity for ongoing research into novel therapeutic targets [10].

One promising direction in HCC research involves the study of oncofetal genes, which are typically expressed during fetal development and re-expressed in various cancers, including HCC [11,12]. These genes play pivotal roles in cancer progression due to their involvement in cell proliferation and vascularization—processes that are crucial both in embryonic development and tumor growth [13]. This similarity has led to the concept of "oncofetal reprogramming," which seeks to understand and exploit the parallels between embryonic development and cancer progression for therapeutic gain [14,15]. Recent studies have shown that reprogramming of oncofetal genes in HCC plays a crucial role in promoting tumor progression and immune evasion [16]. This research highlights the significance of targeting these genes within the tumor microenvironment, suggesting that such an approach could offer a more precise and effective treatment strategy compared to traditional therapies [17,18]. This approach differentiates itself by directly addressing the underlying mechanisms driven by oncofetal genes, offering new potential in the fight against HCC [19,20].

In the context of oncofetal research, the zebrafish (*Danio rerio*) model has emerged as a powerful tool for in vivo studies of liver development and cancer [21]. Zebrafish embryos provide a unique platform for investigating the role of specific genes in liver differentiation and tumorigenesis, facilitating a comprehensive integration of in silico, in vitro, and in vivo methodologies [22]. Recent studies have highlighted the critical role of key transcription factors regulated by signaling molecules in liver development, with zebrafish serving as an ideal model system to unravel these complex interactions [23,24]. Additionally, the zebrafish model has been successfully used to replicate liver tumorigenesis driven by specific oncogenes, offering valuable insights into the molecular pathways involved in hepatocellular carcinoma [25,26]. These findings suggest the versatility and effectiveness of zebrafish as a model organism in advancing our understanding of liver biology and cancer [27].

Solute carrier family 2 member 2 (*SLC2A2*) is known to play a role in glucose uptake and metabolic regulation in various tissues, including pancreatic beta cells and renal tubular cells, thereby supporting cell growth and energy homeostasis. Its

ability to regulate intracellular glucose levels is critical for maintaining cellular functions and promoting anabolic processes, particularly in rapidly growing or differentiating cells. Additionally, *SLC2A2* plays a crucial role during embryogenesis, aiding in cell differentiation and metabolic adaptation at early developmental stages. This *SLC2A2* has also been reported to be involved in the differentiation of liver and HCC progression [28]. Our study aims to identify novel therapeutic targets that can improve the prognosis for HCC patients by investigating the expression and functional dynamics of *SLC2A2* in correlation with liver differentiation markers [29,30]. Through a detailed exploration of *SLC2A2* role in these processes, using a combination of molecular classification techniques and the zebrafish model [31], this research will enhance our understanding of the molecular mechanisms driving HCC progression and contribute to the development of targeted therapies that overcome the limitations of current treatment paradigms [5,32].

## Materials and methods

### Data collection

Three datasets (GSE132606 [33], GSE25417 [34], and GSE67848 [35]) were obtained from the Gene Expression Omnibus database to investigate the relationship between *SLC2A2* expression and stemness over time. Each dataset contained gene expression profiles measured at multiple time points. The fragments per kilobase of transcript per million mapped for liver cancer RNA-sequencing data from The Cancer Genomic Atlas (TCGA) was downloaded from the Genomic Data Commons, and patients with unavailable clinical information were excluded.

### Statistical and bioinformatic analysis

Statistical analyses were performed to assess the significance of the relationship between *SLC2A2* expression and stemness. Correlation analysis and Spearman's correlation coefficients were used to quantify the strength and direction of the association between *SLC2A2* and stemness gene expression levels. Additionally, statistical tests such as analysis of variance or t-tests were conducted to compare the expression levels of *SLC2A2* and stemness markers at different time points or experimental conditions. In R Programming Language software, the R package "ggscatter" was used to correlation analysis.

### Zebrafish maintenance

Wild-type AB zebrafish were obtained from the Korea Zebrafish Core Resource Center (KZRC) and maintained in an automatic circulation system (Genomic-Design) at 28.5°C. All experiments involving zebrafish were cRonducted in accordance with the Institutional Animal Care and Use Committee (IACUC) guidelines of Pusan National University (PNU-2023–0359). Zebrafish embryos used in the experiments were cultured in E3 media (5 mM NaCl, 0.17 mM KCl, 0.4 mM $CaCl_2$, and 0.16 mM $MgSO_4$) in incubators set at 28°C. Zebrafish embryos were sacrificed under deep anesthesia using tricaine at a concentration of 6 g/L, which exceeds the standard anesthetic dose (2 g/L) to ensure rapid and humane euthanasia. This method was chosen to minimize potential suffering.

### Morpholino injection

A splice-blocking morpholino targeting exon3/intron3 of *SLC2A2* (Gene Tools) was dissolved in DEPC water at 25 ng/nL stock. The sequence of morpholino targeting *SLC2A2* is 5′-CAAGTTCACAGATACTCCACCTTCC-3′. A morpholino targeting *SLC2A2* was injected into embryos of wild-type AB zebrafish at the one-cell stage post-fertilization. Microinjections were performed using a FemtoJet 4i microinjector (Eppendorf, Hamburg, Germany).

### RNA isolation and quantitative PCR using zebrafish embryos

To isolate zebrafish embryos, they were homogenized using RNase-free pestles. Total RNA was extracted from the homogenized embryos using 1 mL TRIzol reagent (Molecular Research Center Inc.) manually. Chloroform was added

to separate proteins, and isopropanol was added to precipitate total RNA. Using SuperScript IV Reverse Transcriptase (Thermo Fisher Scientific, Waltham, MA, USA) 3 µg of total RNA was reverse-transcribed. Real-time polymerase chain reaction (RT-PCR) was performed using GoTaq G2 DNA Polymerase (Promega), and the results were visualized using BANDi-Green Nucleic Acid Stain (Translab, Daejeon, Korea). A quantitative RT-PCR (qPCR) assay was performed using PowerUp SYBR Green Master Mix (Thermo Fisher Scientific). The sequence of primers used for RT-qPCR are listed in Table 1. We analyzed the target genes expression levels using the comparative threshold method, and the results were normalized to β-actin as endogenous control.

### Whole-mount *in situ* hybridization (WISH)

WISH was performed on 2- and 4-d post-fertilization (dpf) embryos. The embryos were fixed in 4% paraformaldehyde in phosphate-buffered saline and dehydrated with methanol at -20°C overnight. The samples were incubated with acetone at -20°C for permeabilization and were hybridized with a digoxigenin (DIG)-labeled antisense RNA probe in a hybridization buffer (50% formamide, 5× SSC, 500 µg/mL Torula yeast tRNA, 50 µg/mL heparin, 0.1% Tween-20, and 9 mM citric acid; pH 6.5) for 3 d. The samples were then washed using 2× and 0.2× SSC solutions. The washed samples were blocked with normal goat serum and bovine serum albumin (BSA) and then incubated with alkaline phosphatase-conjugated DIG antibodies (1:5000) (Roche, Basel, Switzerland) overnight at 4°C. The samples were finally incubated with alkaline phosphatase reaction buffer (100 mM Tris; pH 9.5), 50 mM $MgCl_2$, 100 mM NaCl, and 0.1% Tween-20, and the NBT/BCIP substrate (Promega) was used to visualize the WISH signal.

### Statistical analysis of zebrafish experiments

For statistical analysis of the zebrafish experiments, Student's t-test was used, and all experiments were performed in triplicate. The figures and graphs represent averages of three independent experiments. The error bars indicate the standard error of the mean. p-values <0.05 were considered statistically significant.

**Table 1. Primers used for qPCR gene expression analysis.**

| Gene symbol | Forward primer | Reverse primer |
| --- | --- | --- |
| AFP | acatcctcagcttgctgtct | aatgcttggctctcctggat |
| CK19 | ctttgtgtcctcgtcctcct | gtcgcggatcttcacctcta |
| DLK | gcttcatcgacaagacctgc | caggtctcgcacttgttgag |
| EPCAM | cagaaggagatcacaacgcg | tccagatccagttgttcccc |
| GLUT2 | ccagaaagccccagatacctttac | cagcatcagtgccactagaatagg |
| IGF1 | catgtcctcctcgcatctcttcta | atctccagcctccttagatcacag |
| IGF1R | aaaccttcgcctcatcctaggaga | tttatgtccccttctgctttggcgc |
| IGF2 | caatggggaagtcgatgctg | ggaaacagcactcctcaacg |
| Nanog | tgagtgtggatccagcttgt | tctctgcagaagtgggttgt |
| PROM1 | ttcttgaccgactgagaccc | ccaagcacagagggtcattg |
| SALL4 | atttgtgggaccctcgacat | ctgagttattgttcgcccccg |
| Sox2 | tgatggagacggagctgaag | gcttgctgatctccgagttg |
| Sox9 | atgaagatgaccgacgagca | aacttgtcctcctcgctctc |
| GAPDH | catgttcgtcatggggtgaacca | agtgatggcatggactgtggtcat |
| g6pc1a.1 | tcacagcgttgctttcaatc | acttggtgtgggaaatgagc |
| Igf1a | cacactgtccttccccaagt | gatgaccagggcgtagttgt |

## Cell culture

Human HCC cells (HepG2) were purchased from the Korean Cell Line Bank. HepG2 cells were cultured in Dulbecco's modified Eagle medium supplemented with 10% fetal bovine serum (Gibco, CA, USA) at 37°C with 5% $CO_2$.

## Gene expression by lentiviral infection

Lentiviral particles were generated for target gene expression and infection, as described in a previous study [36]. The target sequence was 5′-ACCAATTCCAGCTACCGAC-3′ (sh*SLC2A2*), and 5′- CGAGATCTATGGACTACAAGGACGA CGATGACAAGATGACAGAAGATAAGGTCA-3′ (*SLC2A2* over).

## qPCR using cell line

Total RNA was extracted using a RNeasy Mini Kit (Qiagen, Hilden, Germany). Complementary DNA was synthesized using a Smart Gene Compact cDNA Synthesis Kit (Smart Gene, South Korea). qPCR was performed using the Light-Cycler 96 Real-Time PCR System (Roche). Target mRNA expression relative to the housekeeping gene expression (GAPDH) was calculated using the ΔΔCT method. The sequence of primers used for RT-qPCR are listed in Table 1.

## Western blotting

HepG2 cells were harvested, homogenized in lysis buffer, and centrifuged at 13,000 rpm for 20 min at 4°C. In total, 30 µg of protein underwent 10% sodium dodecyl sulfate-polyacrylamide gel electrophoresis. Proteins were then transferred onto nitrocellulose membranes that were blocked with 5% BSA, and incubated with antibodies against anti-GLUT2 (ABclonal Technology, MA, USA) overnight at 4°C (1:500 dilution). The membranes were then probed with an anti-β-actin antibody (ABclonal Technology) for 3 h (1:1,000 dilution) as an internal control. Antigen-antibody complexes were detected using chemiluminescence (Thermo Fisher Scientific).

## Fluorescent Imaging of Transgenic Zebrafish Embryos

*Fabp10a:mCherry* transgenic zebrafish embryos were used for fluorescent imaging. Embryos from the control group and those injected with *slc2a2*-MO were imaged at 72, 96, and 120 hours post-fertilization (hpf). Prior to imaging, embryos were anesthetized using tricaine (0.016% w/v) and mounted in 3% methylcellulose to ensure stability during imaging. Fluorescent signals were captured using a fluorescent stereo microscope (Zeiss, Discovery V8)dissecting fluorescence microscope equipped with appropriate filters for detecting mCherry fluorescence. Images were acquired under identical conditions for all experimental groups to allow for consistent comparison.

## Immunohistochemistry using zebrafish embryos

Zebrafish embryos were fixed in 4% paraformaldehyde at 4°C overnight, washed twice with PBT (PBS + 0.1% Tween-20), dehydrated through a PBT/MeOH gradient, and incubated in 100% MeOH at -20°C overnight. After rehydration through a MeOH/PBT gradient, samples were incubated in 1% Triton X-100 for 1 hour at room temperature, washed twice with PBT, and blocked for 1 hour at room temperature. Primary antibody incubation was performed with IGF1R antibody (A0243, ABclonal) diluted 1:100 in antibody buffer for 1–3 days at 4°C. Samples were then washed with sodium phosphate buffer and incubated with a 1:100 dilution of secondary antibody in the dark for 1 hour at room temperature. After washing with PBT, samples were imaged using a fluorescent stereo microscope (Zeiss, Discovery V8)

## Statistical analysis

All data were analyzed using Student's t-test, and p-values were used to assess statistical significance as follows: * p <0.05, ** p <0.01, and *** p <0.001.

## Results

### Negative correlation between *SLC2A2* and liver stemness-related genes

We investigated temporal changes in the expression profiles of mature liver markers in the GSE132060, GSE25417, and GSE67848 datasets to validate the data. The expression patterns of albumin, G6PC, and CYP3A4 showed an upward trend in these datasets (Fig 1A). Conversely, we observed a simultaneous decrease in the expression levels of stemness-associated genes NANOG, SOX2, and POU5F1 during liver development (Fig 1B). The GSE132060, GSE25417, and GSE67848 datasets also demonstrated a consistent increase in SLC2A2 expression during development across the three datasets (Fig 1C).

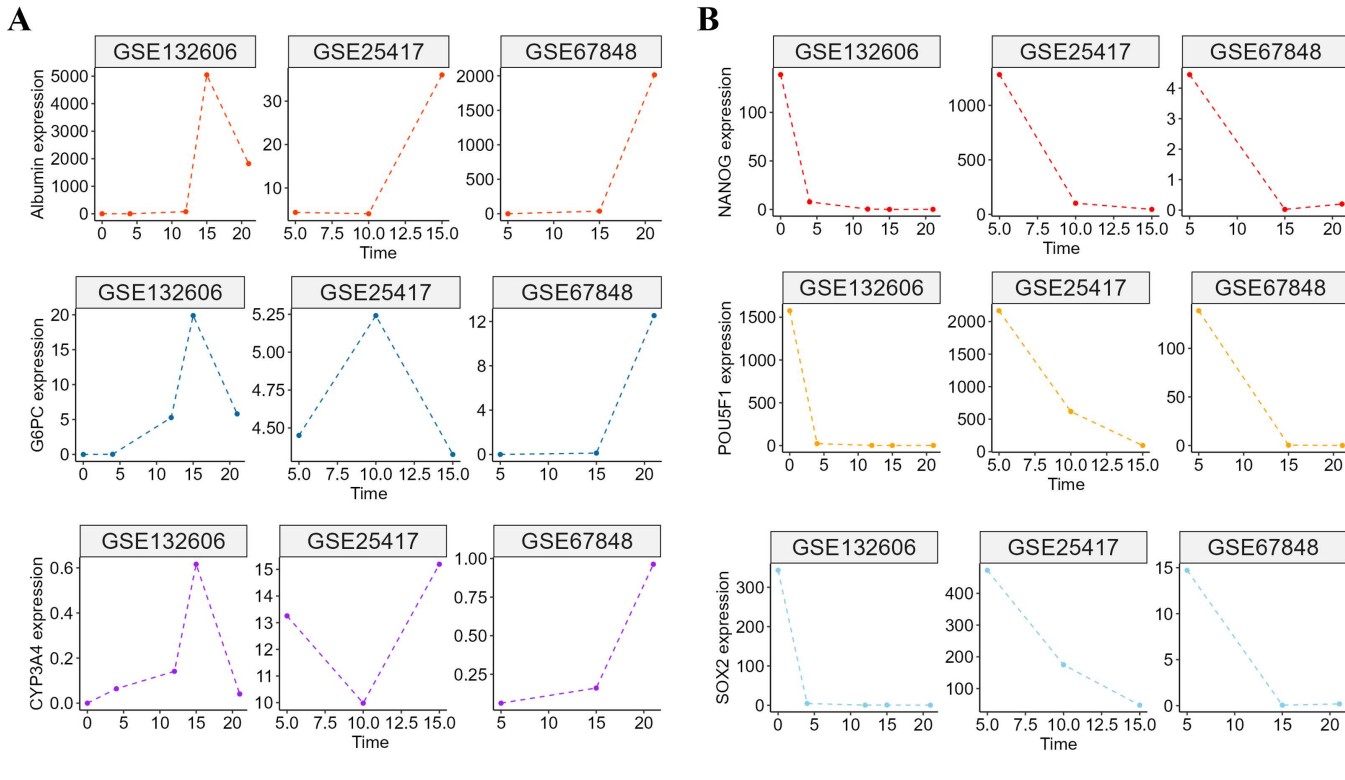

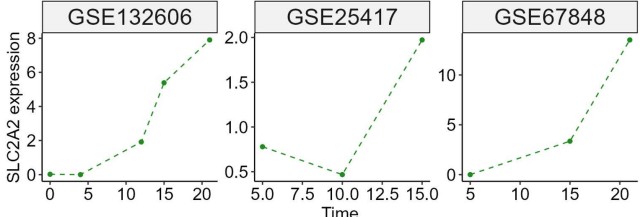

**Fig 1. Gene expression dynamics during liver development.** (A) Gene expression pattern of *albumin, G6PC, and CYP3A4* in GSE132060, GSE25417, and GSE67848 datasets. (B) Gene expression pattern of *NANOG*, *SOX2*, and *POU5F1* in GSE132060, GSE25417, and GSE67848 datasets. (C) Gene expression pattern of *SLC2A2* in GSE132060, GSE25417, and GSE67848 datasets.

## Negative correlation between *SLC2A2* and stemness-related genes

To delineate *SLC2A2* attributes in HCC, we investigated its expression dynamics across different cancer stages using the TCGA HCC dataset. Our analysis revealed a consistent decrease in *SLC2A2* expression with increasing cancer stage. Conversely, *SOX2* and *POU5F1* expression levels progressively increased with advancing cancer stage (Fig 2A). A significant inverse correlation was observed between *SLC2A2* and both stemness-associated (*SOX2* and *POU5F1*) and oncofetal marker genes (*AFP, SALL4, FOXM1*, and *IGF2BP1*) (Fig 2B–2C).

## Knockdown of *SLC2A2* (GLUT2) in HepG2 cells increases the expression of oncofetal and differentiation-related genes

To investigate *SLC2A2* function, *SLC2A2* was knocked down in HepG2. *SLC2A2* knockdown was validated using qRT-PCR (Fig 3A) and western blotting (Fig 3B and S1 Fig). Subsequently, stemness markers were examined after *SLC2A2* genetic regulation in HepG2. Stem cell markers expression, including *AFP, CK19, DLK1, EPCAM, IGF2, PROM1*, and *SALL4*, increased after *SLC2A2* knockdown. In contrast, *SLC2A2* overexpression reduced *PROM1, CK19, DLK1, SALL4* and *EPCAM* expression (Fig 3C).

## *slc2a2* is essential for liver differentiation during vertebrate development

To optimize the morpholino dosage, a series of concentrations were injected into the embryos. Among the different doses tested, 5 ng/embryo was ideal for achieving sufficient *slc2a2* knockdown (Fig 4A). The embryo phenotype was also observed after injecting different doses of morpholino. At a higher dose of 10 ng/embryo, off-target effects such as reduced brain and eyeball size and heart edema were observed at 5 dpf (Fig 4B). Therefore, a 5 ng/embryo dose was chosen as it achieved effective knockdown without inducing these off-target phenotypes.

Subsequently, the expression of liver differentiation markers was examined using whole mount in situ hybridization (WISH) and qPCR. Knock-down of *slc2a2* resulted in the reduced expression of *fabp10a* in the liver at 5 dpf, as observed by WISH (Fig 4C). Additionally, *slc2a2* knockdown decreased the expression of mature hepatocyte markers, including *fabp10a* and *g6pc1a.1*, as shown by qPCR (Fig 4E). However, slc2a2 knockdown did not significantly affect the expression of the hepatoblast marker *hhex* at 2 dpf, as assessed by WISH (Fig 4D) and qPCR (Fig 4F). Representative fluorescence microscopy images and quantitative analysis of liver fluorescence intensity in zebrafish embryos injected with *slc2a2*-MO compared to uninjected controls at 72, 96, and 120 hpf (Fig 4G).

## IGF1R pathway is upregulated following slc2a2 knockdown in HepG2 cells and zebrafish embryos

Knockdown of *slc2a2* using shGLUT2 significantly increased the expression of *IGF1R* and *IGF2* compared to the pLKO control group. In contrast, overexpression of *slc2a2* (GLUT2 over) significantly decreased *IGF1R* and *IGF2* expression compared to the pMSCV control group. Notably, the reduction in *IGF2* expression was confirmed through in vitro qPCR analysis (Fig 5A). RT-qPCR analysis further revealed a significant increase in *IGF1R* mRNA expression in zebrafish embryos following *slc2a2* knockdown (Fig 5B). Additionally, protein level of IGF1R was measured in zebrafish embryos at 2 dpf via immunohistochemistry (IHC). The fluorescence intensity of IGF1R was markedly stronger in the *slc2a2-MO* group compared to the control group, indicating an increase in IGF1R protein expression (Fig 5C).

## Discussion

SLC2A2 (GLUT2), a critical glucose transporter, plays a pivotal role in hepatic glucose metabolism but remains understudied in the context of liver differentiation [37]. This study integrated in silico, in vitro, and in vivo approaches to elucidate the relationship between SLC2A2 expression, liver differentiation, and its broader impact on progression of HCC.

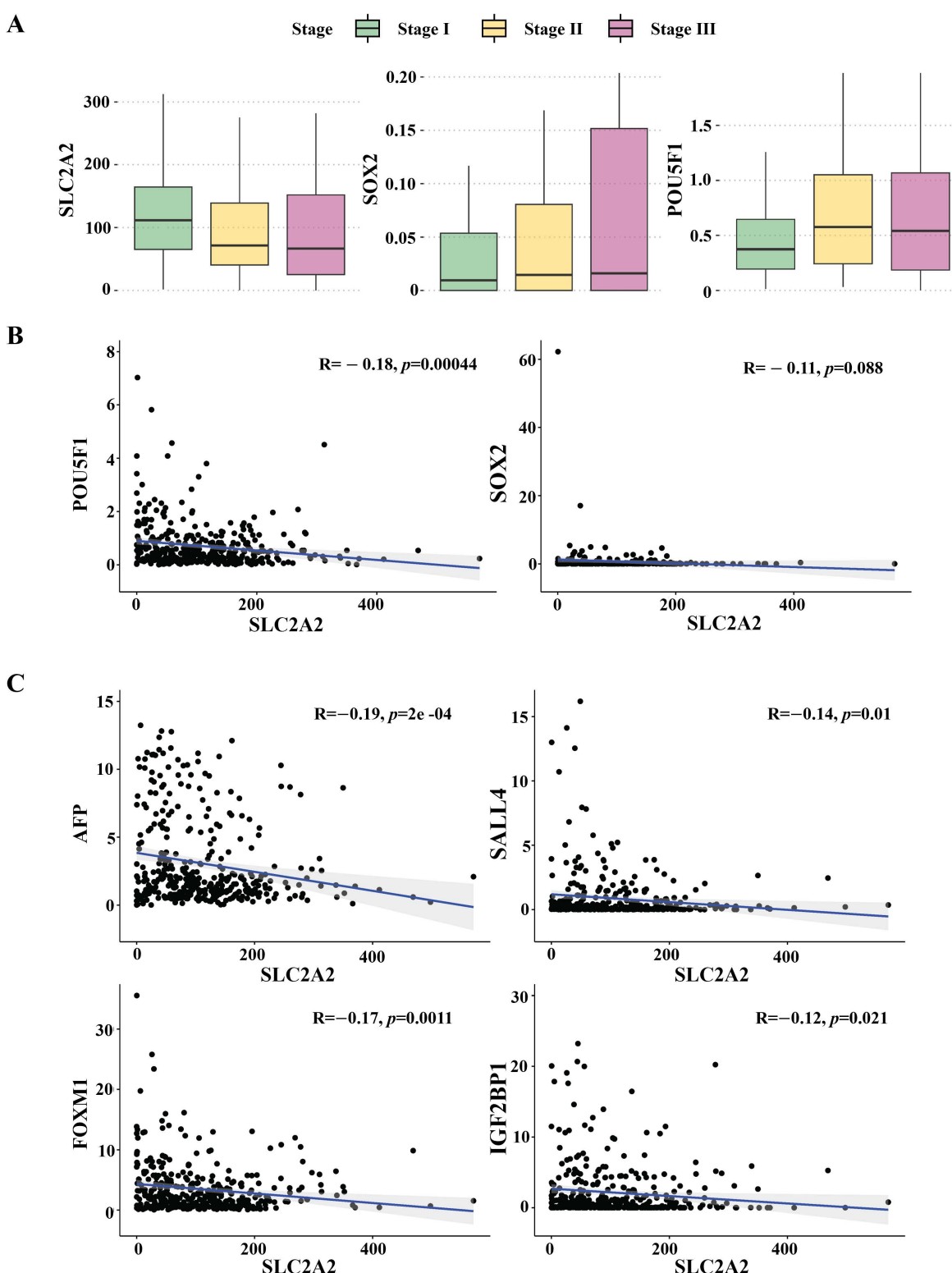

**Fig 2. Changes in *SLC2A2* gene expression in hepatocellular carcinoma (HCC) based on its stage and correlation with oncofetal and stemness genes.** (A) The bar plot illustrates the expression levels of *SLC2A2*, *SOX2*, and *POU5F1* genes in HCC samples at different stages (Stages I, II, and III). (B) The scatter plot depicts the correlation between the expression levels of the *SLC2A2,* stemness (*SOX2* and *POU5F1*), and (C) oncofetal genes (*AFP, SALL4*, *FOXM1*, and *IGF2BP1*). Each data point represents an individual sample, and the correlation coefficients and p-values are indicated.

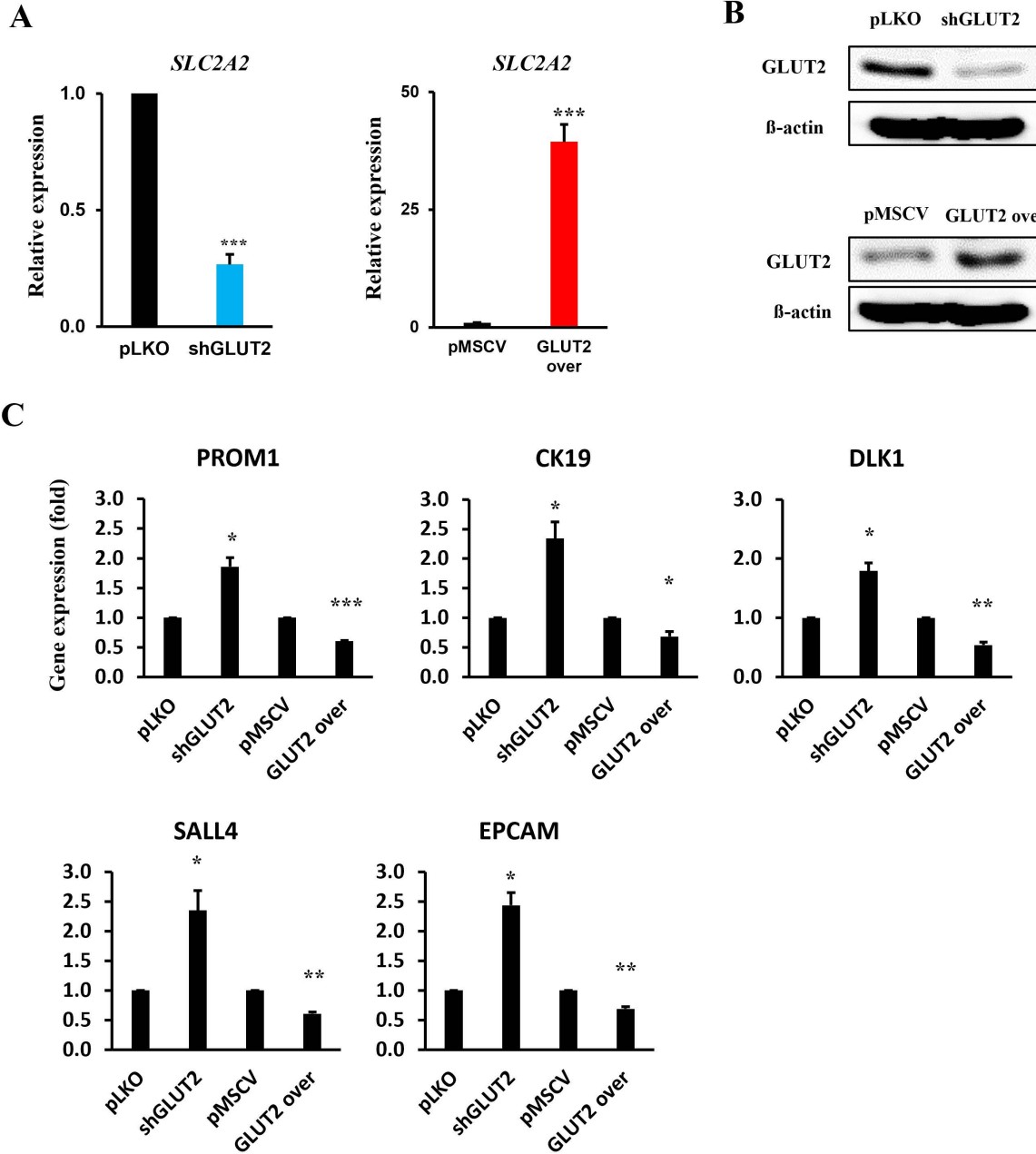

**Fig 3. *SLC2A2* is associated with IGF1R pathways differentiation.** (A) *SLC2A2* expression levels were determined using quantitative real-time polymerase chain reaction (qPCR) in or GLUT2 overexpression cells compared with control cells (n = 4). (B) GLUT2 protein levels were measured by western blotting (n = 3). (C) We confirmed whether the difference in *SLC2A2* expression level was related to stemness (n = 3) or (D) the IGF1R pathways (n = 3) via qPCR. * p <0.05, ** p <0.01, and *** p <0.001.

Our analysis demonstrated that SLC2A2 expression was significantly reduced in HCC compared to other SLC2A family members (Fig 2A). This reduction correlated with increased expression of stem cell markers, such as PROM1, CK19, DLK1, SALL4, and EPCAM (Fig 3C), suggesting that low SLC2A2 levels may promote cancer stem cell traits, malignancy, and therapeutic resistance. These findings align with previous studies showing that diminished SLC2A2 expression correlates with poorer HCC prognosis [28].

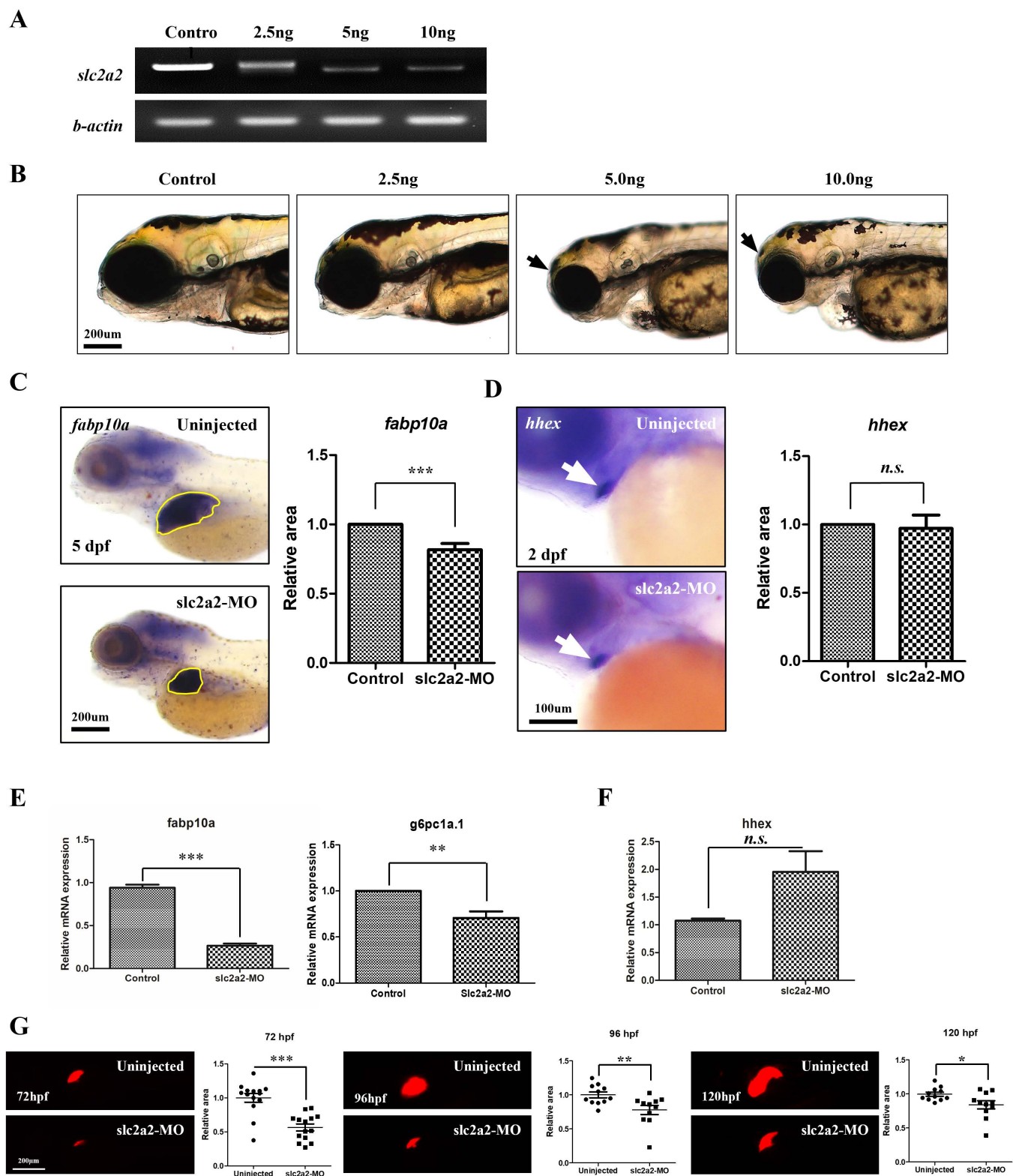

**Fig 4. SLC2A2 is essential for liver differentiation in developing vertebrates** (A) RT-PCR analysis of *SLC2A2* and β-actin using 4 dpf zebrafish embryos. Total RNA was isolated from uninjected control and *SLC2A2* MO-injected embryos (2.5, 5, and 10 ng). (B) Lateral view of zebrafish embryos

after *SLC2A2* MO-injection (2.5, 5, and 10 ng). The black arrow indicates heart edema in zebrafish embryos injected with 10 ng of *SLC2A2*-MO. (C) WISH images of uninjected embryos and *SLC2A2*-targeting morpholino-injected embryos at 5 dpf using *fabp10a*. The yellow line indicates *fabp10a* signal at the liver. (D) Using hepatoblast marker *hhex*, we captured WISH images of uninjected embryos and *SLC2A2*-targeting morpholino-injected embryos at 2 dpf. The white arrow indicates the *hhex* signal in the hepatoblast. (E) qRT-PCR analysis of *fabp10a* and *hhex* expression in uninjected embryos and *SLC2A2*-targeting morpholino-injected embryos at 4 dpf. mRNA expression is normalized to that of β-actin mRNA levels (*** indicates significance at p-value <0.001). Scale bars indicates 200 um. (F) Quantitative RT-PCR was used to measure *hhex* expression, normalized to β-actin mRNA levels. There was no significant difference (n.s.) observed in hhex expression between control and *SLC2A2* MO-injected embryos. Data are represented as mean ± standard error of the mean (SEM). (G) Quantitative RT-PCR analysis was performed to evaluate the expression levels of igf1r. The mRNA expression levels were normalized to β-actin. A significant increase (p-value < 0.001, ***) in igf1r expression was observed in *SLC2A2* MO-injected embryos compared to the control group. Data are presented as mean ± standard error of the mean (SEM). (G) Reduction in liver fluorescence intensity in zebrafish embryos following slc2a2 morpholino (MO) injection. Representative fluorescence microscopy images and quantitative analysis of liver fluorescence intensity in zebrafish embryos at 72, 96, and 120 hours post-fertilization (hpf). Uninjected embryos show consistent fluorescence in the liver across all time points, while embryos injected with slc2a2 MO exhibit a progressive reduction in fluorescence intensity. The fluorescence signal in slc2a2 MO-injected embryos decreased by approximately 60% at 96 hpf and 80% at 120 hpf compared to the uninjected controls.

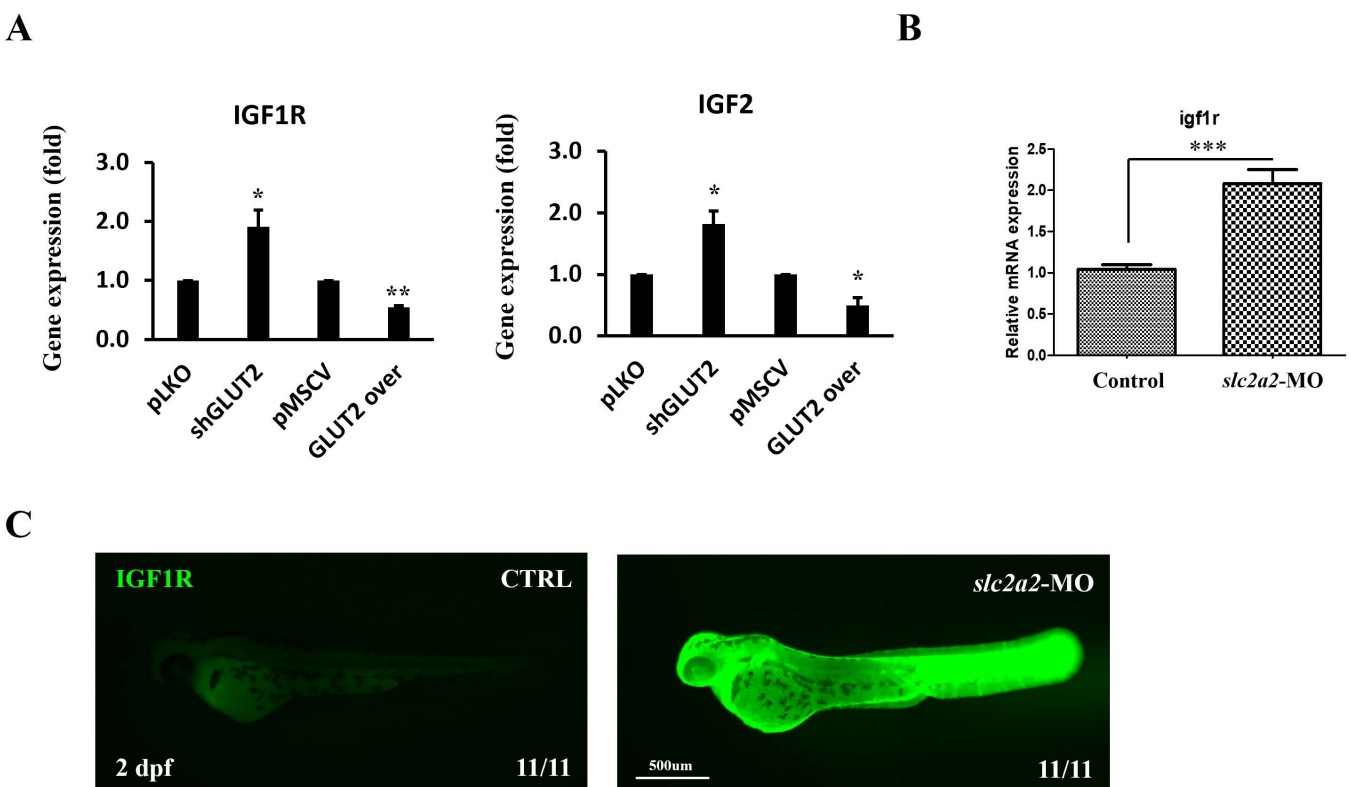

**Fig 5. The IGF1R pathway is upregulated following *slc2a2* knockdown in HepG2 cells and zebrafish embryos.** (A) qPCR analysis of *IGF1R* and *IGF2* gene expression in HepG2 cells. Knockdown of *slc2a2* using shGLUT2 significantly increased *IGF1R* and *IGF2* expression compared to the pLKO control group (*p* < 0.05). Overexpression of *slc2a2* (GLUT2 over) significantly decreased *IGF1R* and *IGF2* expression compared to the pMSCV control group (*p* < 0.01 for *IGF1R*, *p* < 0.05 for *IGF2*). (B) qPCR analysis of *igf1r* mRNA expression in zebrafish embryos at 2 dpf. *slc2a2* knockdown significantly increased *igf1r* expression compared to the control group (*p* < 0.001). (C) Immunohistochemistry (IHC) staining for IGF1R protein in zebrafish embryos at 2 dpf. *slc2a2-MO* treated embryos showed stronger IGF1R fluorescence intensity compared to control embryos, indicating increased IGF1R protein expression. All analyzed embryos (11/11) showed consistent results in both groups.

In addition, our in vitro experiments revealed that SLC2A2 knockdown led to upregulation of IGF1R and IGF2 (Fig 5), key regulators of cell proliferation and survival [38]. This suggests that SLC2A2 may influence liver differentiation by modulating IGF1R signaling pathways. Increased IGF1R expressions likely suppresses differentiation by promoting cellular proliferation, a mechanism commonly observed in cancer biology [39]. This metabolic adaptation may be a compensatory response to glucose scarcity caused by reduced SLC2A2 expression, as similar mechanisms have been reported in other systems using *C. elegans* under starvation stress [38]. Targeting IGF1R in the context of SLC2A2 dysregulation presents a potential therapeutic approach for liver-related diseases, particularly HCC.

Our zebrafish model experiments provided further insights into the role of SLC2A2 in liver development. Morpholino-mediated knockdown of *slc2a2* reduced the expression of *fabp10a*, a marker of mature hepatocytes, without significantly affecting the expression of *hhex*, a hepatoblast marker (Fig 4C–4G). This indicates that while early liver development can proceed independently of SLC2A2, its role becomes increasingly critical as hepatoblasts differentiate into mature hepatocytes. The observed decrease in liver fluorescence intensity and increased IGF1R expression in *slc2a2*-knockdown embryos emphasize its regulatory function in liver differentiation (Fig 5C). The use of zebrafish as a model system is advantageous due to their rapid development, optical transparency, and genetic tractability, which allow for real-time observation of liver differentiation and gene-specific functional studies [40]. These features make zebrafish an ideal platform for investigating complex developmental and disease mechanisms in vivo.

By demonstrating parallels between fetal liver development and HCC progression, this study highlights the utility of zebrafish models in exploring the dual role of SLC2A2 in development and oncogenesis. The re-expression of fetal-like markers in HCC further supports the concept of oncofetal reprogramming, where developmental pathways are hijacked during cancer progression. These findings suggest the therapeutic potential of targeting SLC2A2 to modulate cancer stem cell traits and improve differentiation in HCC. However, further research is required to clarify the molecular mechanisms through which SLC2A2 regulates IGF1R signaling, particularly in the context of liver differentiation and HCC progression. Investigating how SLC2A2-mediated changes in glucose metabolism influence IGF1R pathway activation and downstream effects on cellular proliferation and differentiation will provide deeper insights into its role in liver biology and tumorigenesis.

In conclusion, our study demonstrates that SLC2A2 serves as a critical regulator of liver differentiation and HCC progression. The findings suggest that therapeutic strategies aimed at restoring SLC2A2 expression or counteracting its downstream effects may improve HCC prognosis and provide new avenues for liver-related disease treatments. Future studies should focus on unraveling the precise molecular mechanisms of SLC2A2 in hepatocyte differentiation and its potential as a therapeutic target in liver cancer.

## Conclusions

This study demonstrates that SLC2A2 (GLUT2) plays a critical role in liver differentiation and HCC progression. In silico, in vitro, and in vivo analyses revealed that increased SLC2A2 expression promote hepatocyte differentiation, while its reduction correlates with advanced HCC stages and elevated stem cell markers. These findings highlight SLC2A2 as a potential therapeutic target for improving liver differentiation and suppressing cancer stem cell traits in HCC. Further research is needed to fully elucidate its mechanisms and translate these insights into clinical applications.

## Supporting information

**S1 Fig. Protein levels of GLUT2 were analyzed by western blotting.** GLUT2 protein expression was quantified compared to the control cells., ** $p < 0.01$, and *** $p < 0.001$.
(PPTX)

## Acknowledgments

None

## Author contributions

**Conceptualization:** Yejin Kim, Kyungjae Myung, Ninib Baryawno, Yun Hak Kim, Chang-Kyu Oh.

**Data curation:** Yu Yeuni.

**Formal analysis:** Yu Yeuni, Yun Hak Kim.

**Funding acquisition:** Yun Hak Kim, Chang-Kyu Oh.

**Investigation:** Yu Yeuni, Hye Jin Heo, Eun Sun Kim.

**Methodology:** Hye Jin Heo.

**Project administration:** Yun Hak Kim, Chang-Kyu Oh.

**Resources:** Kyungjae Myung, Ninib Baryawno, Chang-Kyu Oh.

**Supervision:** Yun Hak Kim, Chang-Kyu Oh.

**Visualization:** Eun Sun Kim.

**Writing – original draft:** Yejin Kim.

**Writing – review & editing:** Yun Hak Kim, Chang-Kyu Oh.

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
