## [Decision Letter · Decision Letter 0]

1 Nov 2024

PONE-D-24-37357Solute carrier family 2 member 2 (glucose transporter 2): a common factor of hepatocyte and hepatocellular carcinoma differentiationPLOS ONE

Dear Dr. Kim,

Thank you for submitting your manuscript to PLOS ONE. After careful consideration, we feel that it has merit but does not fully meet PLOS ONE’s publication criteria as it currently stands. Therefore, we invite you to submit a revised version of the manuscript that addresses the points raised during the review process.

**Please address the concerns from the two reviewers.**

We look forward to receiving your revised manuscript.

Kind regards,

Heping Cao, PhD

Academic Editor

PLOS ONE

Journal requirements: When submitting your revision, we need you to address these additional requirements. 1. Please ensure that your manuscript meets PLOS ONE's style requirements, including those for file naming. The PLOS ONE style templates can be found at https://journals.plos.org/plosone/s/file?id=wjVg/PLOSOne_formatting_sample_main_body.pdf and https://journals.plos.org/plosone/s/file?id=ba62/PLOSOne_formatting_sample_title_authors_affiliations.pdf 2. Please match your authorship list in your manuscript file and in the system. 3. To comply with PLOS ONE submissions requirements, in your Methods section, please provide additional information regarding the experiments involving animals and ensure you have included details on (1) methods of sacrifice, (2) methods of anesthesia and/or analgesia, and (3) efforts to alleviate suffering. 4. PLOS ONE now requires that authors provide the original uncropped and unadjusted images underlying all blot or gel results reported in a submission’s figures or Supporting Information files. This policy and the journal’s other requirements for blot/gel reporting and figure preparation are described in detail at https://journals.plos.org/plosone/s/figures#loc-blot-and-gel-reporting-requirements and https://journals.plos.org/plosone/s/figures#loc-preparing-figures-from-image-files. When you submit your revised manuscript, please ensure that your figures adhere fully to these guidelines and provide the original underlying images for all blot or gel data reported in your submission. See the following link for instructions on providing the original image data: https://journals.plos.org/plosone/s/figures#loc-original-images-for-blots-and-gels.   In your cover letter, please note whether your blot/gel image data are in Supporting Information or posted at a public data repository, provide the repository URL if relevant, and provide specific details as to which raw blot/gel images, if any, are not available. Email us at plosone@plos.org if you have any questions. 5. We note that the grant information you provided in the ‘Funding Information’ and ‘Financial Disclosure’ sections do not match.  When you resubmit, please ensure that you provide the correct grant numbers for the awards you received for your study in the ‘Funding Information’ section. 6. Please note that your Data Availability Statement is currently missing [the repository name and/or the DOI/accession number of each dataset OR a direct link to access each database]. If your manuscript is accepted for publication, you will be asked to provide these details on a very short timeline. We therefore suggest that you provide this information now, though we will not hold up the peer review process if you are unable. 7. Please include your full ethics statement in the ‘Methods’ section of your manuscript file. In your statement, please include the full name of the IRB or ethics committee who approved or waived your study, as well as whether or not you obtained informed written or verbal consent. If consent was waived for your study, please include this information in your statement as well.  8. Please include captions for your Supporting Information files at the end of your manuscript, and update any in-text citations to match accordingly. Please see our Supporting Information guidelines for more information: http://journals.plos.org/plosone/s/supporting-information. 

Reviewers' comments:

Reviewer's Responses to Questions

**Comments to the Author**

1. Is the manuscript technically sound, and do the data support the conclusions?

Reviewer #1: Yes

Reviewer #2: Yes

2. Has the statistical analysis been performed appropriately and rigorously? 

Reviewer #1: Yes

Reviewer #2: Yes

3. Have the authors made all data underlying the findings in their manuscript fully available?

Reviewer #1: Yes

Reviewer #2: Yes

4. Is the manuscript presented in an intelligible fashion and written in standard English?

Reviewer #1: Yes

Reviewer #2: Yes

5. Review Comments to the Author

Reviewer #1: This study investigated the effects of GLUT2 on liver differentiation using in vivo, in vitro, and zebrafish models, in conjunction with big data analysis. Analysis of three datasets (GSE132606, GSE25417, GSE67848) from the Gene Expression Omnibus (GEO) database revealed high expression levels of GLUT2 in patients with hepatocellular carcinoma (HCC). However, I still have related questions that I would like to have explained.

1.The introduction of the role of the SLC2A2 gene needs more description and literature support.

2.Source channels for zebrafish purchases need to be supplemented.

3. The clarity of the images needs to be improved.

4.After knocking out the target gene, the PCR results of other indicators did not change significantly.

5.The article has less experimental content, and it is recommended to add relevant protein experiments.

Reviewer #2: This work uses in vitro, in vivo (zebrafish), and in silico models to examine the role of SLC2A2 in hepatic differentiation and hepatocellular cancer (HCC). According to the findings, SLC2A2 expression and genes linked to stemness are negatively correlated, indicating that it plays a critical role in liver cell differentiation. Targeting SLC2A2 in HCC has therapeutic potential, which is further investigated in this study. Several comments will improve the quality of this study.

Major comments.

1. To better establish whether Slc2a2 influences hepatocyte differentiation, it would be beneficial to analyze additional liver-specific markers. This could provide stronger evidence for the role of Slc2a2 in the differentiation process.

2. The use of zebrafish embryos to explore liver differentiation is an excellent idea. As you have examined the relationship between EGFR, IGF1R, and slc2a2 in in vitro models, it would be interesting to investigate whether similar results are observed in the developing embryos.

3. In Figure 1, it would be helpful to check whether well-established liver differentiation markers (such as albumin, CYP3A4, G6PC) increase as the embryos develop. This would help confirm that the datasets from GEO accurately represent time points in embryonic development.

4. While the study has identified that SLC2A2 plays a role in liver differentiation and cancer progression, it does not fully elucidate the underlying mechanisms behind these effects. This lack of mechanistic clarity represents a limitation of the study. I suggest adding this point to the Discussion section of the manuscript to acknowledge this limitation.

Minor comment.

1. Figures 4C and 4D, please provide a more detailed explanation of what the y-axis represents in Figures 4C and 4D, so that the reader can better understand the data.

6. PLOS authors have the option to publish the peer review history of their article (what does this mean? ). If published, this will include your full peer review and any attached files.

**Do you want your identity to be public for this peer review?** For information about this choice, including consent withdrawal, please see our Privacy Policy .

Reviewer #1: No

Reviewer #2: No

---

## [Author Response · Author response to Decision Letter 1]

21 Jan 2025

Point-to-Point Answers to Reviewers Comments

Reviewer #1

1.The introduction of the role of the SLC2A2 gene needs more description and literature support.

Thank you for your kind comments. As you suggested, we have elaborated on the function of slc2a2 in the Introduction section to help readers better understand the manuscript. Below is the additional contents (Please see the attached response letter file).

2.Source channels for zebrafish purchases need to be supplemented.

As per your suggestion, we have made the necessary adjustments. In the Materials and Methods section, we have added that the zebrafish and fabp10a:mCherry transgenic zebrafish embryos used in this study were obtained from the Korea Zebrafish Resource Center (KZRC). Thank you for your valuable input.

3. The clarity of the images needs to be improved.

We agree with your opinion. Accordingly, we have replaced the image with a higher-resolution version to enhance its clarity and quality.

4.After knocking out the target gene, the PCR results of other indicators did not change significantly.

We agree with your opinion. To prevent confusion for the readers, we have revised the figure to display only the differentially expressed genes, removing the remaining genes from the figure.

5.The article has less experimental content, and it is recommended to add relevant protein experiments.

As you suggested, additional experiments seem necessary. Therefore, we utilized the fabp10a:mCherry transgenic zebrafish embryos, which specifically mark the liver, to investigate the effects of slc2a2 knockdown (K/D). Our results showed that the liver size decreased when slc2a2 was knocked down, as confirmed at the protein level (Figure 4G). Additionally, to determine whether IGF1R protein levels also increase upon slc2a2 knockdown, we performed immunohistochemistry (IHC). The IHC results demonstrated that IGF1R expression increased across the whole body when slc2a2 was knocked down (Figure 5C).

Reviewer 2

Major comments.

1. To better establish whether Slc2a2 influences hepatocyte differentiation, it would be beneficial to analyze additional liver-specific markers. This could provide stronger evidence for the role of Slc2a2 in the differentiation process.

Thank you for your kind suggestion. Based on your comments, we measured the mRNA level of g6pc1a.1, another mature liver marker, in zebrafish embryos. Additionally, we analyzed the mRNA levels of albumin, G6PC, and CYP3A4 using public data. Below are the experimental results: (Please see the attached response letter file)

Figure 1A

Figure 4E

2. The use of zebrafish embryos to explore liver differentiation is an excellent idea. As you have examined the relationship between EGFR, IGF1R, and slc2a2 in in vitro models, it would be interesting to investigate whether similar results are observed in the developing embryos.

We agree with your opinion and have conducted experiments to investigate this relationship in zebrafish embryos. After performing a knockdown of slc2a2, we observed an increase in the mRNA level of IGF1R, as confirmed by qPCR. Additionally, IHC analysis revealed an increase in the protein level of IGF1R (Figure 5 B and C). These results suggest that slc2a2 may play a regulatory role in modulating IGF1R expression during liver differentiation in zebrafish embryos. Notably, we also examined EGFR expression but did not observe significant changes in either mRNA or protein levels following slc2a2 knockdown. Therefore, our analysis focused on IGF1R, as it demonstrated a more consistent and significant response. These findings further highlight the importance of IGF1R as a key regulatory pathway influenced by slc2a2 in liver differentiation.

3. In Figure 1, it would be helpful to check whether well-established liver differentiation markers (such as albumin, CYP3A4, G6PC) increase as the embryos develop. This would help confirm that the datasets from GEO accurately represent time points in embryonic development.

Thank you for your kind comments. As per your suggestion, we analyzed the expression levels of albumin, G6PC, and CYP3A4 in datasets GSE132606, GSE25417, and GSE67848. The results demonstrated that the expression of all three genes increased as liver differentiation progressed (Figure 1A).

4. While the study has identified that SLC2A2 plays a role in liver differentiation and cancer progression, it does not fully elucidate the underlying mechanisms behind these effects. This lack of mechanistic clarity represents a limitation of the study. I suggest adding this point to the Discussion section of the manuscript to acknowledge this limitation.

Thank you for your valuable suggestion. As you recommended, we have included this point in the Discussion section of the manuscript to acknowledge the limitation regarding the lack of mechanistic clarity behind the effects of SLC2A2 on liver differentiation and cancer progression. This addition highlights the need for further studies to fully elucidate the underlying mechanisms.

---

## [Decision Letter · Decision Letter 1]

28 Feb 2025

Solute carrier family 2 member 2 (glucose transporter 2): a common factor of hepatocyte and hepatocellular carcinoma differentiation

PONE-D-24-37357R1

Dear Dr. Kim,

We’re pleased to inform you that your manuscript has been judged scientifically suitable for publication and will be formally accepted for publication once it meets all outstanding technical requirements.

Kind regards,

Heping Cao, PhD

Academic Editor

PLOS ONE

Additional Editor Comments (optional):

Reviewers' comments:

Reviewer's Responses to Questions

**Comments to the Author**

1. If the authors have adequately addressed your comments raised in a previous round of review and you feel that this manuscript is now acceptable for publication, you may indicate that here to bypass the “Comments to the Author” section, enter your conflict of interest statement in the “Confidential to Editor” section, and submit your "Accept" recommendation.

Reviewer #2: All comments have been addressed

2. Is the manuscript technically sound, and do the data support the conclusions?

Reviewer #2: Yes

3. Has the statistical analysis been performed appropriately and rigorously? 

Reviewer #2: Yes

4. Have the authors made all data underlying the findings in their manuscript fully available?

Reviewer #2: Yes

5. Is the manuscript presented in an intelligible fashion and written in standard English?

Reviewer #2: Yes

6. Review Comments to the Author

Reviewer #2: The authors have adequately addressed my comments raised in a previous manuscript. So there is no more comment.

7. PLOS authors have the option to publish the peer review history of their article (what does this mean? ). If published, this will include your full peer review and any attached files.

**Do you want your identity to be public for this peer review?** For information about this choice, including consent withdrawal, please see our Privacy Policy .

Reviewer #2: No

---

## [Editor Report · Acceptance letter]

PONE-D-24-37357R1

PLOS ONE

Dear Dr. Kim,

I'm pleased to inform you that your manuscript has been deemed suitable for publication in PLOS ONE. Congratulations! Your manuscript is now being handed over to our production team.

Kind regards,

on behalf of

Dr. Heping Cao

Academic Editor

PLOS ONE